# Circulating Galectin-3 in Patients with Cardiogenic Shock Complicating Acute Myocardial Infarction Treated with Mild Hypothermia: A Biomarker Sub-Study of the SHOCK-COOL Trial

**DOI:** 10.3390/jcm11237168

**Published:** 2022-12-02

**Authors:** Wenke Cheng, Georg Fuernau, Steffen Desch, Anne Freund, Hans-Josef Feistritzer, Janine Pöss, Christian Besler, Philipp Lurz, Petra Büttner, Holger Thiele

**Affiliations:** 1Department of Internal Medicine/Cardiology, Heart Center Leipzig at University of Leipzig, 04289 Leipzig, Germany; 2Medical Faculty, University of Leipzig, 04103 Leipzig, Germany; 3Clinic for Internal Medicine II (Cardiology, Angiology, Diabetology, Intensive Care Medicine), Dessau Community General Hospital, 06847 Dessau-Rosslau, Germany

**Keywords:** Galectin-3, cardiogenic shock, acute myocardial infarction, mild therapeutic hypothermia

## Abstract

Background: Galectin-3 (Gal-3) is considered a potential cardiovascular inflammatory marker that may provide additional risk stratification for patients with acute heart failure. It is unknown whether mild therapeutic hypothermia (MTH) impacts Gal-3 levels. Therefore, this biomarker study aimed to investigate the effect of MTH on Gal-3. Methods: In the randomized SHOCK-COOL trial, 40 patients with cardiogenic shock (CS) complicating acute myocardial infraction (AMI) were randomly assigned to the MTH (33 °C) or control group in a 1:1 ratio. Blood samples were collected on the day of admission/day 1, day 2, and day 3. Gal-3 level kinetics throughout these time points were compared between the MTH and control groups. Additionally, potential correlations between Gal-3 and clinical patient characteristics were assessed. Multiple imputations were performed to account for missing data. Results: In the control group, Gal-3 levels were significantly lower on day 3 than on day 1 (day 1 vs. day 3: 3.84 [IQR 2.04–13.3] vs. 1.79 [IQR 1.23–3.50] ng/mL; *p* = 0.049). Gal-3 levels were not significantly different on any day between the MTH and control groups (*p* for interaction = 0.242). Spearman’s rank correlation test showed no significant correlation between Gal-3 levels and sex, age, smoking, body mass index (BMI), and levels of creatine kinase-MB, creatine kinase, C-reactive protein, creatinine, and white blood cell counts (all *p* > 0.05). Patients with lower Gal-3 levels on the first day after admission demonstrated a higher risk of all-cause mortality at 30 days (hazard ratio, 2.67; 95% CI, 1.11–6.42; *p* = 0.029). In addition, Gal-3 levels on day 1 had a good predictive value for 30-day all-cause mortality with an area under the receiver operating characteristic curve of 0.696 (95% CI: 0.513–0.879), with an optimal cut-off point of less than 3651 pg/mL. Conclusions: MTH has no effect on Gal-3 levels in patients with CS complicating AMI compared to the control group. In addition, Gal-3 is a relatively stable biomarker, independent of age, sex, and BMI, and Gal-3 levels at admission might predict the risk of 30-day all-cause mortality.

## 1. Introduction

Acute myocardial infarction (AMI) is the most common cause of cardiogenic shock (CS), accounting for more than 80% of cases [1]. Early coronary artery reperfusion is the most effective treatment [2]. Despite advances in medication and devices supporting patients with CS, overall mortality has not significantly improved during the past 20 years [3] and is still as high as 50% [4]. CS complicating AMI is a systemic clinical syndrome accompanied by a pronounced systemic inflammatory reaction and severe perfusion insufficiency of multiple organs [5]. Inflammation causes elevated levels of circulating inflammatory cytokines, chemokines, and cell adhesion molecules as well as activation of peripheral leukocytes and platelets; and these alterations result in tissue ischemia, apoptosis, neurohormonal activation, and extracellular matrix degradation [5,6].

The randomized SHOCK-COOL trial has shown that mild therapeutic hypothermia (MTH) does not improve hemodynamic parameters in patients with CS complicating AMI [7]. Nevertheless, MTH has been reported to alleviate the inflammatory response in vitro and in animal models [8,9,10,11]. 

Galectin-3 (Gal-3), a 29–35 kDa protein, is a member of the β-galactoside-binding lectin family [12]. Although not cardiac-specific, Gal-3 is expressed in cardiac cells and has emerged as an important regulator of physiological and pathological processes, including inflammation and fibrosis [13], and has been associated with myocardial infarction and myocardial fibrosis [14]. Moreover, Gal-3 is currently considered a cardiovascular inflammatory marker that can be used to predict the prognosis of patients with heart failure and coronary artery disease [15,16]. Furthermore, the American Heart Association (AHA) guidelines suggest that Gal-3 may be used for additional risk stratification in patients with heart failure [17]. The present biomarker study aimed to elucidate the effect of MTH on Gal-3 in patients with CS as the most severe form of acute decompensated heart failure.

## 2. Methods

### 2.1. Patients and Study Design

The present study is a sub-analysis of the SHOCK-COOL trial (ClinicalTrials.gov (accessed on 1 December 2022) Identifier NCT01890317). The main study analyzed the hemodynamic effects of MTH on the cardiac power index in patients with CS complicating AMI, and the results have been fully published [7]. Briefly, in the SHOCK-COOL trial, 40 patients with CS complicating AMI were recruited at the Heart Center Leipzig at University of Leipzig between July 2012 and March 2015 and were assigned to the MTH (33 °C) or control group using a web-based randomized system in a 1:1 ratio. Patients in the MTH group had cooling initiated in the catheterization laboratory with cooled saline and maintained after percutaneous coronary intervention by a commercially available system (CoolGard^®^, ZOLL Medical Corp, Chelmsford, MA, USA). Central temperature measurements were taken in the urinary bladder, maintained for 24 h after reaching the target temperature (33 °C), and then rewarmed to 37 °C at a rate of 0.25 °C/h. Patients in the control group were treated according to standard care without MTH. All patients underwent early revascularization with percutaneous coronary intervention and mechanical ventilation. The exclusion criteria for the study were as follows: (1) patients aged >90 years; (2) patients with MTH indications for out-of-hospital resuscitation; (3) patients with mechanical complications after AMI; and (4) patients with CS lasting for more than 12 h. The trial was approved by the Local Ethics Committee (Medical Faculty, University Leipzig, registration number 230-12-21052012) and was conducted in compliance with the principles of the Declaration of Helsinki. The process for written informed consent from all patients has been described previously [7].

### 2.2. Laboratory Measurements

Blood samples were collected on day 1 (at admission), day 2, and day 3 under standard conditions. The samples were then centrifuged at 4 °C for 10 min at 1000× *g* to obtain serum and plasma and were subsequently stored in aliquots at −80 °C for future use. The biochemical parameters including creatinine, creatine kinase (CK), creatine kinase-myocardial band (CK-MB), C-reactive protein (CRP), and white blood cell counts were measured by standardized laboratory procedures, and Gal-3 was measured using a commercial enzyme-linked immunosorbent assay (ELISA) kit (R&D systems, Minneapolis, MN, USA). All samples were assayed in duplicate. 

### 2.3. Statistical Analysis

Most variables showed skewed distributions. The continuous variables were expressed as medians and interquartile ranges (IQR), and categorical variables were expressed as counts and proportions. Inter-group differences were analyzed using Fisher’s exact test for dichotomous variables and the Mann–Whitney *U* test for continuous variables. Correlations between Gal-3 levels and different clinical characteristics and biomarkers were analyzed by Spearman’s rank correlation test. The main aim of this study was to compare the differences in Gal-3 levels between and within the MTH and control groups in the first three days following admission. A mixed linear model with random intercepts was employed to fit the data for differences in Gal-3 levels between the MTH and control group within three days after admission with treatment modality as a factor. Furthermore, the values were adjusted for patient characteristics (age, BMI, CK-MB, CRP, CK, white blood cell counts, and creatinine), and time was included as a continuous variable. Differences in Gal-3 levels between the two groups were expressed as medians and 95% confidence intervals (CIs), which were calculated using the Hodges–Lehmann estimator method. Differences in Gal-3 levels within three days in the MTH and control groups were analyzed by nonparametric Kruskal–Wallis and Dunn’s tests. In addition, we performed a sensitivity analysis to account for missing Gal-3 data on days 1, 2, and 3 by multiple imputation, which was based on five replications of the predictive mean matching algorithm and the Markov Chain Monte Carlo method [18]. The secondary outcome was all-cause mortality after 30 days. Regardless of the treatment modality, samples from day 1, 2, and 3 were divided into two groups based on the Gal-3 median, respectively, and time-to-death was estimated by the Kaplan–Meier method and analyzed using the log-rank test. Moreover, the predictive value of Gal-3 for 30-day all-cause mortality was analyzed by the area under the receiver operating characteristic (ROC) curve. Statistical analyses were performed using GraphPad Prism (version 9.0; San Diego, CA, USA), STATA (Version 12.0; Stata Corporation, College Station, TX, USA), and SPSS (version 26; SPSS Inc., Chicago, IL, USA) software. A two-tailed *p*-value < 0.05 was considered statistically significant.

## 3. Results

In the SHOCK-COOL trial, 40 patients aged 50–87 years (median, 76 years) were enrolled at the Heart Center Leipzig and randomly assigned to MTH or control, with 20 patients in each group. From these patients, 38 blood samples from day 1 were available for testing, 31 from day 2, and 25 from day 3 (Figure 1). Ultimately, after quality control, 38 samples (19 MTH and 19 control) on day 1, 30 samples (14 MTH and 16 control) on day 2, and 25 samples (11 MTH and 14 control) on day 3 were included in the data analyses. The demographic and clinical characteristics, except CK, of patients in the two groups within three days after admission were comparable (Table 1).

### Gal-3 Levels in MTH and Control Groups

The Gal-3 levels between the MTH and control groups were comparable during the three days after admission (Figure 2). There were no significant differences in Gal-3 levels between the MTH and control groups over time (*p* for interaction = 0.242, Figure 2); day 1 (MTH 3.08 [IQR 1.45–6.40] vs. control 3.84 [IQR 2.04–13.3] ng/mL; median difference, −1.34 ng/mL; 95% CI, −4.67 to 1.14; *p =* 0.223; Appendix A), day 2 (MTH 2.39 [IQR 1.22–4.39] vs. control 3.0 [IQR 1.38–6.09] ng/mL; median difference, −0.37 ng/mL; 95% CI, −2.50 to 1.26; *p =* 0.532), and day 3 (MTH 2.24 [IQR 0.97–4.38] vs. control 1.79 [IQR 1.23–3.50] ng/mL; median difference, 0.27 ng/mL; 95% CI, −1.31 to 1.74; *p =* 0.770). Sensitivity analyses were performed by multiple imputation of missing data, and the results were consistent before and after imputation (Appendix A). In the control group, Gal-3 levels were higher on day 1 than on day 3 (day 1 vs. day 3:3.84 [IQR 2.04–13.3] vs. 1.79 [IQR 1.23–3.50] ng/mL; mean rank difference, 12.06; *p* = 0.049; Appendix A), yet this was not observed in the MTH group (*p >* 0.05; Appendix A). These results remained consistent before and after imputation. Gal-3 levels within three days after admission were not correlated with sex, age, smoking, BMI, as well as the levels of CK-MB, CK, CRP, creatinine, and white blood cell count (*p >* 0.05 for all).

Furthermore, as a secondary outcome, we explored the potential association between median Gal-3 levels at day 1, 2, and 3 and 30-day all-cause mortality. Patients with lower Gal-3 levels on the first day after admission showed a higher risk of all-cause mortality at 30 days (hazard ratio, 2.67; 95% CI, 1.11–6.42; *p* = 0.029; Figure 3), regardless of treatment modality. Gal-3 levels on days 2 and 3 were not associated with 30-day all-cause mortality (day 2, hazard ratio, 1.81; 95% CI, 0.61–5.35; *p* = 0.285; day 3, hazard ratio, 1.09; 95% CI, 0.27–4.42; *p* = 0.903). Moreover, the ROC curve depicted a good predictive value of Gal-3 levels on day 1 for 30-day all-cause mortality, and the under-area ROC curve (AUC) was 0.696 (95% CI: 0.513-0.879), as shown in Figure 4. The optimal cut-off point was less than 3651 pg/mL, with a sensitivity and specificity of 72.7% and 81.3%, respectively.

## 4. Discussion

This is the first study analyzing the effects of MTH on Gal-3 levels and kinetics during three days after admission in patients with CS complicating AMI. The findings of this study are summarized as follows: (1) The levels of Gal-3 in the control group gradually decreased during three days after admission. (2) MTH had no effect on Gal-3 levels compared to the control group. (3) Regardless of treatment modality, higher Gal-3 levels on day 1 may be associated with a lower risk of 30-day all-cause mortality. Furthermore, Gal-3 levels at admission had a good predictive value for the risk of all-cause mortality at 30 days. (4) Gal-3 levels are independent of age, sex, and BMI in patients with CS complicating AMI.

MTH has been reported to reduce mortality and neurological damage after cardiac arrest [19] and has been recommended for patients with out-of-hospital cardiac arrest caused by ventricular fibrillation [20]. However, randomized trials on hypothermia in cardiac arrest often excluded patients with CS. Therefore, our knowledge of the effect of MTH on these patients remains limited.

Gal-3 is primarily secreted by activated macrophages [21] and has been identified as an initiating molecule involved in the pathogenesis of various diseases characterized by tissue injury and/or stress [22,23]. Gal-3 is released during the acute phase of AMI and reflected by its elevated levels in peripheral blood [24]. Indeed, in the control group, Gal-3 levels were highest on the day of admission and decreased significantly until day 3. Bivona et al. observed comparable Gal-3 patterns in AMI patients for 5 days, presumed an association with revascularization, and proposed a pharmacological intervention mechanism [25].

MTH may have several beneficial effects in patients with CS complicating AMI: (i) reduced metabolic rate and increased left ventricular contractility without increasing oxygen consumption [26,27]; (ii) improved post-ischemic cardiac function and reduced myocardial injury [28]; (iii) reduced end-organ damage due to prolonged hypoperfusion [29]; and (iv) reduced inflammatory response due to hypothermia, as suggested by in vitro models [8,9,10,11]. Because of these observations, we hypothesized that MTH may regulate the inflammatory response and thus decrease the circulating levels of Gal-3 [29]. However, in the present study, this was not the case. This finding is consistent with the proteomic findings of Mohammad et al. who analyzed cardiovascular and inflammatory biomarkers in patients with ST-segment elevation myocardial infarction treated with hypothermia versus normothermic controls [30]. The exact mechanism of the effect of MTH on Gal-3 is unknown, but it has been hypothesized that activated macrophages and other inflammatory cells express Gal-3 under various pathological conditions such as tissue injury [31]. Interestingly, previous studies have reported that Gal-3 expression in brain tissue was inhibited by hypothermia [32], whereas Gal-3 release in breast tissue was not affected by temperature [33]. Thus, the degree of tissue injury and the intensity of the inflammatory response are expected to correlate with Gal-3 levels. On the other hand, as shown by Mohammad et al., the effect of hypothermia on biomarker peaks was modest, so the impact of MTH on Gal-3 may be limited [30]. The present study analyzed the circulating levels of Gal-3, which are representative of total Gal-3 levels; thus, the origin of Gal-3 is unclear, and specific tissue and organ damage cannot be assessed. Unlike control group, Gal-3 levels in the MTH group were comparable over the three days, presumably as a result of the joint effect of hypothermia and rewarming. Indeed, a growing number of studies has demonstrated that hypothermia and rewarming can have pro- or anti-inflammatory effects, which are closely associated with the target temperature of hypothermia, the duration of hypothermia, the rate of rewarming, and the activation of complement after rewarming [30,34,35,36].

Currently, there is strong evidence that Gal-3 is not a simple bystander but an important player in the cardiac remodeling process after AMI [24]. During cardiac injury, cardiomyocytes release cytokines such as tumor necrosis factor-α, interleukin (IL)-18, IL-6, and IL-1b, which subsequently activate macrophages, which rapidly enhance the expression and release of Gal-3 [37]. Upregulation of Gal-3 plays a crucial role in the initial phase of tissue repair [38] and has anti-apoptotic activity rescuing cardiomyocytes, thus reducing myocardial infarct size in vivo [39,40]. However, Gal-3 overexpression interferes with the early stages of cell death, mediated by the perturbation of mitochondrial homeostasis and the formation of reactive oxygen species [41]. Therefore, depending on the amount of the initial Gal-3 increase and the kinetics over time, Gal-3 may be protective or destructive. These underlying mechanisms may explain the higher risk of mortality within 30 days in patients with low Gal-3 levels on day 1.

Over the past decade, several novel biomarkers have been identified in patients with CS complicated by AMI. Growth-differentiation factor 15 and catalytic iron levels on admission predict short-term mortality risk in patients with CS complicating AMI [42,43]. High levels of angiopoietin-2 are independently associated with increased short- and long-term mortality risk [44]. Similarly, we have previously shown that monocyte chemoattractant protein-1 levels on admission may be associated with short-term mortality [45]. However, these studies all suffer from limitations such as single-center, inadequate sample size, and small number of events. Hence, more studies are needed for their future validation. It is worth noting that Gal-3 has been reported to be a stable biomarker independent of age, BMI, and sex [16], which is consistent with our observations. Moreover, Gal-3 has a half-life of a few hours [46] and is stable for nine freeze-thaw cycles after storage at −20 °C or −70 °C [47]. These properties of Gal-3 make it a good candidate for possible clinical applications in the future.

## 5. Limitations

This study has some limitations. First, this was a single-center study with Caucasian patients only and a high proportion of patients with old age and obesity; therefore, the generalizability of the results may be limited. Second, the small sample size is insufficient to adjust for additional variables (e.g., troponin, left ventricular ejection fraction, etc.) in the regression analysis; therefore, no conclusions about their effect on mortality could be drawn. Due to the absence of previous relevant studies as references, a post hoc analysis of statistical efficacy was performed, and a power of 82% for the current results was calculated. The presented analysis of short-term mortality was only exploratory and cannot adequately address these prognostic questions. Third, patients with early-stage CS have a very high mortality rate, and hence missing data is common in these patients. To ensure maximum statistical efficiency, multiple imputations were performed to account for the missing data. Although the complete case analysis yielded consistent results, potential bias due to missing data cannot be ruled out. Fourth, patients underwent echocardiography, but no detailed data in addition to left ventricular ejection fraction were documented in the case report form. Although the hemodynamics of the MTH and control groups were comparable throughout the study, the influence of inter-patient differences in cardiac structure and function on the results of this study cannot be excluded.

## 6. Conclusions

In conclusion, MTH has no effect on Gal-3 levels in patients with CS complicating AMI. In addition, Gal-3 is a relatively stable biomarker, independent of age, sex, and body mass index, and Gal-3 levels at admission might predict the risk of 30-day all-cause mortality.

## Figures and Tables

**Figure 1 jcm-11-07168-f001:**
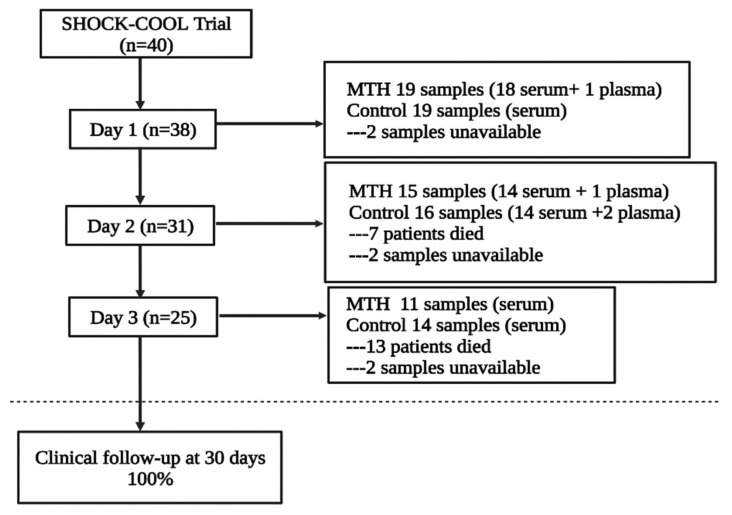
Study flow. SHOCK-COOL, the randomized trial of mild hypothermia for cardiogenic shock. MTH, mild therapeutic hypothermia.

**Figure 2 jcm-11-07168-f002:**
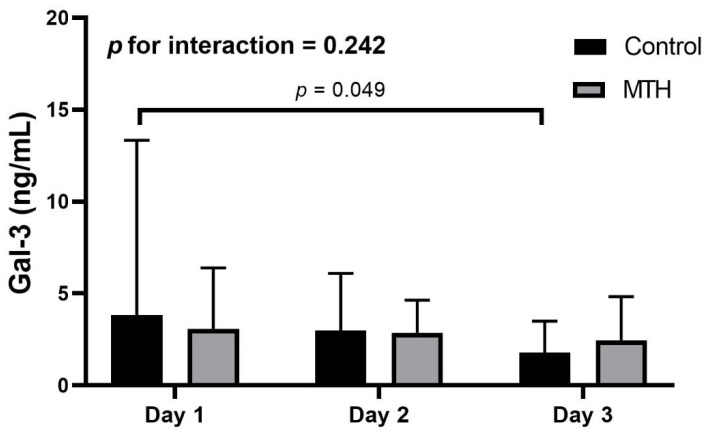
Galectin-3 levels of patients in cardiogenic shock after acute myocardial infarction treated by either MTH or control groups during three days after admission. Box and whisker plots show median and interquartile range. MTH, mild therapeutic hypothermia.

**Figure 3 jcm-11-07168-f003:**
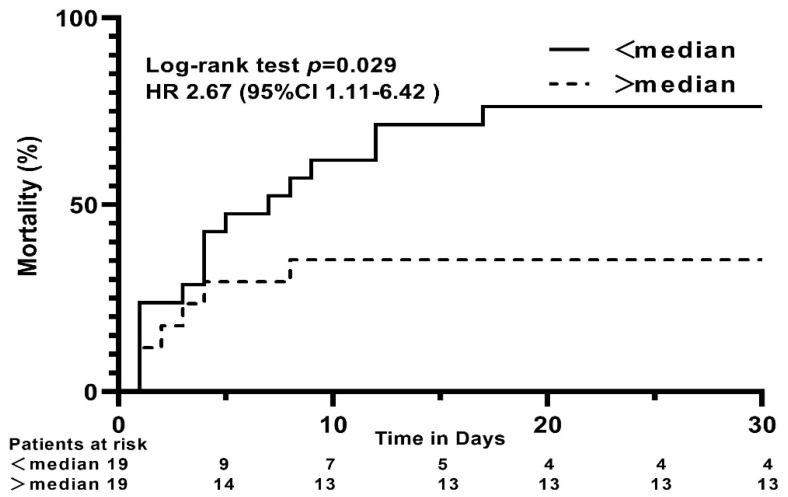
Kaplan–Meier analysis for all-cause mortality at 30 days in cardiogenic shock complicating acute myocardial infarction patients with Galectin-3 levels <median (black solid line) and >median (black dashed line) on day 1.

**Figure 4 jcm-11-07168-f004:**
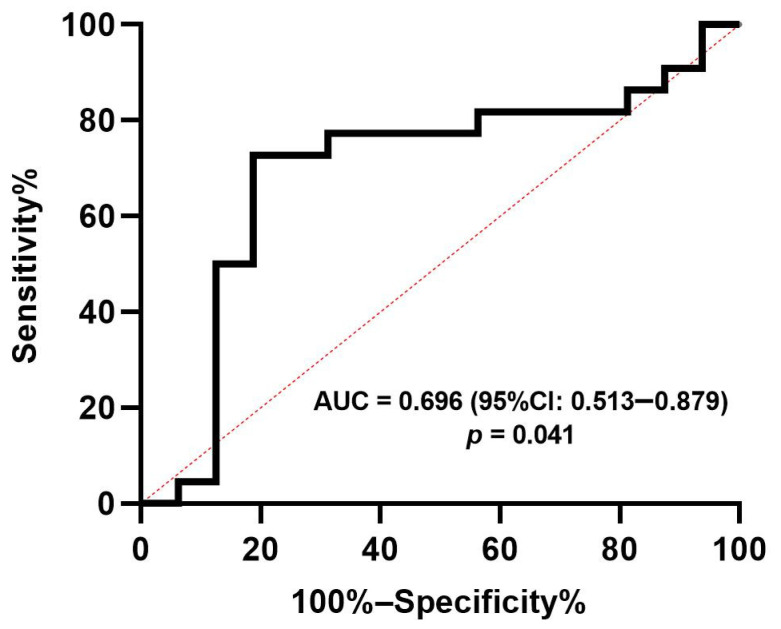
Predictive value of Galectin-3 levels on day 1 (at admission) for 30-day all-cause mortality in patients with CS complicating AMI. The under-area ROC curve was 0.696 (95% CI: 0.513–0.879). The optimal cut-off point was less than 3651 pg/mL, with sensitivity and specificity of 72.7% and 81.3%, respectively. The red dotted line indicates the diagonal line of the ROC curve. CS, cardiogenic shock. AMI, acute myocardial infraction.

**Table 1 jcm-11-07168-t001:** Demographic and clinical characteristics of patients during three days of hospitalization.

	Day 1			Day 2			Day 3		
	MTH (*n* = 19)	Control (*n* = 19)	*p*-Value	MTH (*n* = 14)	Control (*n* = 16)	*p*-Value	MTH (*n* = 11)	Control (*n* = 14)	*p*-Value
Age, years (IQR)	76 (71–80)	75 (70–81)	0.79	76 (68.5–78.5)	75.5 (70.8–82)	0.67	75 (61–77)	73.5 (65–81.5)	0.68
Female, *n* (%)	8 (42.1)	5 (26.3)	0.5	4 (28.6)	5 (31.3)	>0.99	5 (45.5)	5 (35.7)	0.7
Active smoker, *n* (%)	4 (21)	4 (21)	>0.99	4 (28.6)	3 (18.8)	0.67	2 (18.2)	4 (28.6)	0.66
Diabetes mellitus, *n* (%)	7 (36.8)	4 (21)	0.48	4 (28.6)	3 (18.8)	0.67	4 (36.4)	4 (28.6)	>0.99
CK (U/L) (IQR)	6.4 (3–21.9)	19.7 (7.4–39)	0.1	4.8 (1.8–15)	19.1 (9.8–35.3)	0.022	3.8 (2.8–9.2)	15 (4.7–29)	0.067
BMI, kg/m^2^ (IQR)	27.5 (23.6–33)	27.8 (26–31)	0.85	26.9 (23.5–30.6)	27.8 (25–30.7)	0.53	29.4 (23–36.7)	28.4 (25.7–31)	0.85
White blood cell, 10^9^/L (IQR)	16.5 (13–19)	13 (9.6–17.7)	0.12	15.9 (13.2–18.2)	14.1 (10.5–17.8)	0.3	16.8 (15.2–19.3)	13.1 (8.5–17.5)	0.1
Creatinine, µmol/L (IQR)	131 (78–217)	158 (99–250)	0.26	123.5 (69–171)	147 (95–223)	0.22	119 (67–162)	146.5 (91.5–227)	0.32
CRP (IQR)	21.4 (3.1–72)	12.4 (7–64.9)	0.72	5.5 (2.5–78.6)	14.3 (7.2–64.3)	0.48	12 (2.4–86.1)	14.3 (7.6–63.1)	>0.99
Maximum CK-MB, U/L(IQR)	2.6 (1.4–6.2)	3.5 (1.9–8.8)	0.49	2.3 (1.3–8.5)	2.6 (1.8–9)	0.71	2.7 (5.51–1.41)	2.64 (1.6–6.6)	0.85

MTH, mild therapeutic hypothermia. IQR, interquartile range. CK, creatine kinase. CK-MB, creatine kinase-MB. CRP, C-reactive protein.

## Data Availability

Not Applicable.

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
