# Peer review of "Circulating Galectin-3 in Patients with Cardiogenic Shock Complicating Acute Myocardial Infarction Treated with Mild Hypothermia: A Biomarker Sub-Study of the SHOCK-COOL Trial"

_jcm, 2022, doi:10.3390/jcm11237168_

Round 1

Reviewer 1 Report

Circulating Galectin-3 in Patients with Cardiogenic Shock Complicating Acute Myocardial Infarction Treated with Mild Hypothermia: A Biomarker Sub-study of the SHOCK-COOL Trial.  The authors investigated the effect of MTH on Gal-3. Certainly, the study of new potential biomarkers for risk stratification of patients with cardiogenic shock is a promising direction. Results are clearly exposed and well written. However, I must mind some other details that set the work not suitable to publish in its actual state. I would like to explain some changes that I guess will improve the quality of the paper:

1.       Line 22. …SHOCK-COOL trial, 40 patients with CS complicating AMI were randomly assigned to MTH (33℃)… Please, provide an abbreviation decoding for CS and AMI.

2.       Please, provide a power calculation for your study.

3.       The optimal cut-off point of Gal-3 was less or more than 3651 pg/ml?

4.       Do you have any Echo data?

5.       Have the authors compared the significance of galectin-3 levels with other best-studied predictors of adverse outcomes, such as troponin, LVEF, etc.?

6.       Have the authors revealed any articles that were published in 2021-2022?

But in general, I think that this is a very worthy work. I express my gratitude to the authors for their work and my great pleasure in reading their results.

Author Response

Response to Reviewer 1 Comments

Circulating Galectin-3 in Patients with Cardiogenic Shock Complicating Acute Myocardial Infarction Treated with Mild Hypothermia: A Biomarker Sub-study of the SHOCK-COOL Trial.  The authors investigated the effect of MTH on Gal-3. Certainly, the study of new potential biomarkers for risk stratification of patients with cardiogenic shock is a promising direction. Results are clearly exposed and well written.

Response: Thanks for the reviewer's affirmation and positive comments.

However, I must mind some other details that set the work not suitable to publish in its actual state. I would like to explain some changes that I guess will improve the quality of the paper:

Point 1: Line 22. …SHOCK-COOL trial, 40 patients with CS complicating AMI were randomly assigned to MTH (33℃)… Please, provide an abbreviation decoding for CS and AMI.

Response 1: Thanks for the reviewer's comments. This detail has been modified. See line 22-23 in the revised manuscript.

Point 2: Please, provide a power calculation for your study.

Response 2: Thanks for the reviewer's comments. We have added a post-hoc analysis of statistical power to the limitation section. The current power of the manuscript data is 82%. See line 270-272 in the revised manuscript.

Point 3: The optimal cut-off point of Gal-3 was less or more than 3651 pg/ml?

Response 3: Thanks for the reviewer's comments. The optimal cut-off point of Gal-3 was less than 3651 pg/ml. The corresponding parts have been revised.

Point 4: Do you have any Echo data?

Response 4: All patients underwent percutaneous coronary intervention and were randomized immediately after admission. Patients underwent echocardiography but no detailed data in addition to left ventricular ejection fraction were documented in the case report form. However, the main study has investigated the effect of MTH on cardiac power index (CPI). In addition, the CPI levels at baseline and during treatment were comparable in the MTH and control groups. Therefore, there were no significant differences in hemodynamics between the two patient groups throughout the study. We have reported this shortcoming in the limitations section. See line 277-281.

Point 5: Have the authors compared the significance of galectin-3 levels with other best-studied predictors of adverse outcomes, such as troponin, LVEF, etc.?

Response 5: Thanks for the reviewer's valuable suggestions. As stated in this paper, survival analysis was performed as a secondary outcome. Therefore, the predictive potential of galectin-3 was only estimated as an exploratory analysis. The small sample size may not be sufficient to adjust outcomes for more variables in regression analysis. Therefore, we did not include troponin, LVEF, or performed other additional analyses. This is stated in the limitations section. See line 267-269.

Based on your suggestions, this section was expanded accordingly. We have briefly summarized and compared several novel biomarkers that were recently examined in patients with CS complicating AMI. See line 250-258.

Point 6: Have the authors revealed any articles that were published in 2021-2022?

Response 6: The current manuscript is the first submission. Previously, we have measured another biomarker (Monocyte Chemoattractant Protein-1) that has been published in the Journal of Cardiovascular Development and Disease (PMID: 36005444). The results of previous study have also been described. See line 254-255.

But in general, I think that this is a very worthy work. I express my gratitude to the authors for their work and my great pleasure in reading their results.

Response: Thanks again to the reviewer for the positive comments. Your valuable suggestions have greatly improved the readability of the manuscript.

Reviewer 2 Report

The authors sought to investigate the effect of MTH on Galectin-3 in patients with CS complicating AMI in the pooled data of SHOCK-COOL trial. The authors showed the characteristics of Galectin-3 in patients with AMI complicating CS. Although Gal-3 is a relatively stable biomarker, independent of age, sex, and BMI, and Gal-3 levels at admission can predict the risk of all-cause mortality, MTH has no effect on Gal-3 levels in patients with AMI complicating CS compared to control group.

The present study reported the negative data about Gal-3. The authors did not directly show the mechanism why the beneficial impacts of MTH for patients with AMI complicated CS did not link to the serum level of Gal-3. The authors should present why MTH did not influence Gal-3, and why MTH exerted the beneficial impacts for patients with CS.

Author Response

Response to Reviewer 2 Comments

The authors sought to investigate the effect of MTH on Galectin-3 in patients with CS complicating AMI in the pooled data of SHOCK-COOL trial. The authors showed the characteristics of Galectin-3 in patients with AMI complicating CS. Although Gal-3 is a relatively stable biomarker, independent of age, sex, and BMI, and Gal-3 levels at admission can predict the risk of all-cause mortality, MTH has no effect on Gal-3 levels in patients with AMI complicating CS compared to control group.

The present study reported the negative data about Gal-3. The authors did not directly show the mechanism why the beneficial impacts of MTH for patients with AMI complicated CS did not link to the serum level of Gal-3. The authors should present why MTH did not influence Gal-3, and why MTH exerted the beneficial impacts for patients with CS.

Response: Thanks for the reviewer's comments. Your valuable suggestions have greatly improved the readability of the manuscript.

According to your comments, we have provided an appropriate explanation as to why MTH has no effect on Gal-3 and why MTH may have beneficial effects on CS patients. See line 212-217 and 222-232.

Round 2

Reviewer 1 Report

The authors have adressed all my comments and improved much the article.